# A Finite Element Method Study on a Simulation of the Thermal Behaviour of Four Methods for the Restoration of Class II Cavities

**DOI:** 10.3390/jfb15040086

**Published:** 2024-03-30

**Authors:** Adela Nicoleta Staicu, Mihaela Jana Țuculină, Cristian Niky Cumpătă, Ana Maria Rîcă, Maria Cristina Beznă, Dragoș Laurențiu Popa, Alexandru Dan Popescu, Oana Andreea Diaconu

**Affiliations:** 1Department of Endodontics, Faculty of Dentistry, University of Medicine and Pharmacy of Craiova, 200349 Craiova, Romania; cenusoiu.adela@yahoo.com (A.N.S.); r_ana_maria22@yahoo.com (A.M.R.); alexandrudanpopescu20@gmail.com (A.D.P.); oanamihailescu76@yahoo.com (O.A.D.); 2Department of maxillofacial surgery, Faculty of Dental Medicine, University Titu Maiorescu of Bucharest, 67A Gheorghe Petrascu Str., 031593 Bucharest, Romania; 3Department of Pathophysiology, Faculty of Dentistry, University of Medicine and Pharmacy of Craiova, 200349 Craiova, Romania; bezna.mariacristina@gmail.com; 4Department of Automotive, Transportation and Industrial Engineering, Faculty of Mechanics, University of Craiova, 200478 Craiova, Romania; popadragoslaurentiu@yahoo.com

**Keywords:** class II cavity, dental restoration, thermal behaviour, finishing and polishing, dental pulp

## Abstract

The possibility of dental pulp damage during dental procedures is well known. According to studies, during finishing and polishing without cooling, temperatures of up to 140 °C or more can be generated. There are many studies that have analysed the influence of the finishing and polishing of fillings on the mechanical parameters, but the analysis of thermal parameters has led to uncertain results due to the difficulty of performing this in vivo. **Background**: We set out to conduct a study, using the finite element method, to determine the extent to which the type of class II cavity and the volume of the composite filling influence the duration of heat transfer to the pulp during finishing and polishing without cooling. **Materials and Methods**: A virtual model of an upper primary molar was used, with a caries process located on the distal aspect, in which four types of cavities were digitally prepared: direct access, horizontal slot, vertical slot and occlusal–proximal. All four cavity types were filled using a Filtek Supreme XT nanocomposite. **Results**: The study showed that the filling volume almost inversely proportionally influences the time at which the dental pulp reaches the critical temperature of irreversible damage. The lowest duration occurred in occlusal–distal restorations and the highest in direct access restorations. **Conclusions:** based on the results of the study, a working protocol can be issued so that finishing and polishing restorations without cooling are safe for pulpal health.

## 1. Introduction

In the last decades, due to increasing aesthetic demands from patients, important improvements have been made to composite resins, allowing them to be used also in the restoration of posterior teeth [1]. Composite resins are the direct restorative materials of choice for dentists, with good success rates and long-term clinical performance [2]. Annual failure rates vary from 1% to 5% for anterior teeth and 1% to 3% for posterior teeth, respectively [3]. While, in the past, the use of an amalgam required the creation of retentive cavities, its replacement with composite resins has allowed a more conservative approach [4]. Nowadays, it is suggested that cavity design is dictated solely by the extension of the caries process, thanks to the strong adhesive bond between the composite and the remaining hard tissues. This consideration is in accordance with biological principles, but also those of a mechanical nature [5]. Therefore, in cases of the existence of carious processes on the proximal faces of the posterior teeth, in addition to the classic cavities, cassette (in the case of direct access) and compound occlusal–proximal (in the case of indirect access), new forms of preparation have emerged: tunnel cavity, vertical slot and horizontal slot [6,7].

Any direct or indirect restorative material must have tooth-like physico-mechanical properties in order to function in the oral cavity for as long as possible [8]. Both teeth and restorative materials are subject to conditions in the oral cavity such as humidity, temperature variations, occlusal forces, plaque and other external influences, making it necessary to find materials that can withstand these demands [9].

The literature includes many studies aimed at analysing the behaviour of composite resins to mechanical [10,11], chemical and thermal stresses [12] but, in the case of the latter, few studies have also addressed the influence of the effect on the dental pulp of heat occurring during the finishing and polishing of composites.

The requirements expected from composite resins are: increased mechanical strength, reduced thermal conductivity and structural stability. These properties can be provided by modifying or improving the inorganic filler [13,14]. Therefore, the thermal conductivity of composite resins is one of the most important characteristics [15], especially since maintaining dental pulp homeostasis through correct treatment is the main goal of restorative dentistry [16].

The thermal parameters of dental filling materials should be as close as possible to the parameters of human teeth, which are poor thermal conductors [17].

Although studies claim that composite resins have low thermal conductivity [18,19], the results of studies that have examined thermal effects on the pulp are difficult to quantify and compare, given the multitude of variables included (thermal source, intensity, duration of action, etc.) [20].

Although heat transfer in human teeth is a process that occurs every day in everyday life, of great clinical importance is the temperature generated during dental treatments. There are various stages during dental treatments that can generate elevated intrapulpal temperatures: removal of altered tissues using the handpiece at high speeds, exothermic reactions occurring during the light-curing of composite materials, and the polishing and finishing stage of fillings [21]. C. M. Hannah and G. A. Smith conducted infrared tomographic evaluations in their study and found that the surface temperature of the composite subjected to finishing and polishing without cooling was at least 140 °C [19,22].

In addition to this, the heat produced during finishing and dry polishing affects the interface between tooth and adhesive, as well as the bond between the particles and the surrounding matrix [19].

There is a lack of clear knowledge about the actual amount of heat transfer that occurs during dental procedures. This is all the more important as we are referring to a pulp organ already affected by various injuries [21]. Given that intrapulpal temperature recording is an invasive and infeasible process, performing in vitro simulations of intrapulpal temperature changes occurring in various clinical situations becomes the most optimal option.

Therefore, we decided to carry out an in vitro study, using the finite element method, in order to investigate to what extent the volume of the cavity influences the transmission of the heat generated during the finishing and polishing of the fillings to the dental pulp.

The finite element method allows safe, fast and relatively inexpensive simulation, and is increasingly used in restorative dentistry [23,24]. The finite element method has been used to analyse the temperatures occurring intrapulpally during the finishing and polishing of fillings made in an upper permanent first molar. We investigated the extent to which cavity design and obturation volume influence temperature transmission to the pulp, thus simulating four second-class cavities: occlusal–proximal, vertical slot, horizontal slot and box-type cavities.

The null hypothesis of the study (h0) was that all four types of filled cavities equally transmit the temperature that occurs during the finishing and polishing stage of the composite.

## 2. Materials and Methods

### 2.1. Dental Material

To obtain the virtual models underlying the finite element method analysis, an upper permanent first molar was used, which had a carious cavity on the distal surface. It was subjected to two main three-dimensional scanning operations, before and after restoration. This restorative operation was performed using the Filtek Supreme XT nanocomposite produced by 3M ESPE Germany (Seefeld, Germany).

### 2.2. Hardware and Equipment

To determine the different geometries of the upper primary molar under analysis, the 3DSYSTEMS CAPTURE 3D scanner was used. Two computer systems were used to process the data, models and simulations, a laptop computer with an INTEL Core I5 processor 2.6 GHz and RAM 16 GB, and a desktop computer with an INTEL Core I3 processor 3.7 GHz and RAM 8 GB, both made in the USA.

### 2.3. Software

Geomagic for SolidWorks (3D Systems, Rock Hill, SC, USA) [25] was used for 3D scanning and primary processing.

SolidWorks (Dassault Systèmes, Velizy-Villacoublay, France), a computer-aided design (CAD) program, was used to transform surface geometries into virtual solids and prepare them for finite element analysis.

Ansys Workbench (Ansys, Inc., Canonsburg, PA, USA) is a finite element method (FEM) program [26,27,28] which was used for the thermal analyses of the upper permanent first molar models.

The Microsoft Office package was used for comparative analyses of the results.

### 2.4. Work Method

The following methods were used in this study: direct engineering methods (Direct Engineering); computer-aided design methods (CAD); reverse-engineering methods (Reverse Engineering); the finite element method (Finite Element Method—FEM); thermodynamic-specific methods; and dental cavity restoration methods and techniques (Direct Access, H Slot, Distal–Occlusal and V Slot). The thermal properties of the materials used were integrated into the calculation algorithm of the Ansys program based on the principles and laws of thermodynamics.

### 2.5. Virtual Models of the Upper Permanent First Molar Specific to Restorative Techniques 

#### 2.5.1. Virtual Model of the Upper Permanent First Molar with Carious Cavity

The 3DSYSTEMS CAPTURE 3D scanner and Geomagic for SolidWorks software (both components produced in the USA) were used to obtain a primary geometry of the molar. Some of the specific 3D scanning operations are shown in Figure 1. Scan alignment techniques were used.

A model containing 52,806 triangular primary surfaces was obtained after successive scans. This model was subjected to techniques of refinement, decimation of the number of triangular surfaces and removal of non-conforming surfaces. Finally, the model was imported into SolidWorks and transformed into a virtual solid.

#### 2.5.2. Virtual Exterior Model of the Upper Permanent First Molar

It was also necessary to obtain the integral molar model. For this, reverse engineering techniques were used in the Geomagic software. Thus, the cavity model was subjected to elimination and then filling operations with tangent surfaces. The final model was loaded into SolidWorks and transformed into a virtual solid. These transformations are shown in Figure 2.

#### 2.5.3. External Virtual Models of Dentin and Pulp of the Upper Permanent First Molar

To determine the geometry of molar dentin, full or partial offset techniques were applied to the whole tooth model. Similarly, to obtain the pulp model, offset techniques were applied successively to the whole model or to selections of it.

#### 2.5.4. Modelling of Direct Access, Occlusal–Distal, Vertical Slot and Horizontal Slot External Cavities

Using CAD methods and techniques, direct access, occlusal–distal, vertical slot and horizontal slot cavities were modelled on the cavity model as shown in Figure 3.

#### 2.5.5. Virtual Model of the Intact Upper Permanent First Molar

In order to obtain the complete molar model, the outer molar model, the dentin model and the pulp model were loaded into the Assembly module of SolidWorks. Since the models come from the outer molar model, the three models have the same coordinate system. For this reason, the three reference planes were used for the correct alignment of the models. Figure 4 shows the alignment of the three outer models.

#### 2.5.6. Virtual Model of Composite Fillings Used in Cavity Restoration

In order to obtain the composite filling model for the direct access cavity, the outer models of the intact molar and the direct access cavity molar were loaded into SolidWorks, as shown in Figure 5 (the outer model of the intact molar is coloured green).

The two models have been aligned based on reference planes that are common because they come from the same primary model.

By volume subtraction, the composite layer model is obtained in the direct access cavity situation. A similar procedure was followed in obtaining composite obturation patterns for the occlusal–distal, vertical slot and horizontal slot cavities, as shown in Figure 6.

#### 2.5.7. Virtual Models of the Restored Molar for Direct Access, Occlusal–Distal, Vertical Slot and Horizontal Slot Cavities

In order to obtain these models, the full model of the analysed whole molar was loaded into the Assembly module of SolidWorks. In turn, the composite models for the four cavity types were loaded and aligned. Using volume subtraction for each situation, the complete models for the four cavity types were obtained. Figure 7 shows the final models that were subjected to thermal analysis later.

### 2.6. Simulation of the Thermal Behaviour of the Virtual Models of the Molar Analysed after Cavity Restoration in the Finishing and Polishing Process

#### 2.6.1. Common Conditions

An important issue that arises when defining a simulation of this type is given by the amount of specific intraoral thermal convection. Thus, Haiwei X. and collaborators, in 2017, determined the convection phenomenon, using experimental and virtual techniques, as having values ranging from 2 W/m^2^ × K to 3 W/m^2^ × K [29]. In these simulations, it was considered to be 3 W/m^2^ × K. Also, analysing the friction phenomenon that exists between the finishing and polishing tools and the composite, we proposed a thermal protocol for the outer surfaces of the cavity-restoration layer [19,22]. Thus, it was considered that the temperature of these surfaces increases from 37 °C to 140 °C in 5 s and then remains at 140 °C for another 10 s. In total, the thermal process analysed took 15 s and is described by the diagram in Figure 8.

Since the systems analysed have not static but transient behaviour, Transient Thermal was chosen as the simulation type in Ansys Workbench.

The initial temperature of the systems analysed was considered to be 37 °C. Also, the structure of the materials composing these systems is very important, especially their thermal behaviour. The materials are introduced in the simulations with their physico-thermal characteristics. Thus, Table 1 shows the materials used and their properties, which are synthesized from the analysis of several papers [30,31,32,33,34,35,36,37].

#### 2.6.2. Simulation of the Thermal Behaviour of the Virtual Models of the Molar Analysed after Direct Access Cavity Restoration in the Finishing and Polishing Process

The analysed model for the direct access restoration was imported into Ansys Workbench. The model was divided into tetrahedral finite elements, resulting in a structure consisting of 1,057,356 nodes and 663,797 finite elements, as shown in Figure 9.

The surface to which the finishing process was applied was the outer surface of the composite layer. The temperature source was applied to this surface for 15 s, as shown in Figure 10. The convection phenomenon that occurs on contact with air in the oral cavity was applied to the free surfaces of the molar, as can be seen in Figure 10.

#### 2.6.3. Simulation of the Thermal Behaviour of the Virtual Models of the Molar Analysed after Occlusal–Distal Cavity Restoration

The molar model to which the occlusal–distal cavity restoration was applied was imported into Ansys Workbench. The Transient Thermal module was chosen, which can simulate transient thermal situations.

In the next stage, the model was subjected to the tetrahedron-splitting operation, the structure having 627,556 nodes and 388,852 finite elements. The outer surface of the composite becomes, in Ansys Workbench, the source of the temperature acting on the system since, in reality, it is subjected to the finishing operation. The thermal convection phenomenon is also present in the intraoral cavity. 

#### 2.6.4. Simulation of the Thermal Behaviour of the Virtual Models of the Molar Analysed after Restoration of the Vertical Slot Cavity in the Finishing and Polishing Process

The geometrical model of the vertical slot cavity restoration system was imported into Ansys Workbench. This model was divided into finite elements of the tetrahedron type, resulting in 606,613 nodes and 376,055 elements. The external surfaces of the composite were selected as the temperature source for the system under analysis. The convection phenomenon was created in Ansys Workbench by selecting the free surfaces of the molar under analysis.

#### 2.6.5. Simulation of the Thermal Behaviour of the Virtual Models of the Molar Analysed after Restoration of the Horizontal Slot Cavity in the Finishing and Polishing Process

The molar model with horizontal slot cavity restoration was imported into Ansys Workbench in the Transient Thermal module, which can analyse transient thermal regimes. In the next step, the model was split into tetrahedron-like finite elements, yielding 1,185,070 nodes and 749,134 elements. The temperature source was placed on the outer surfaces of the composite. The thermal convection phenomenon is present on the free surfaces of the molar. For this reason, convection was carried out on the selected surfaces of the anatomic crown.

## 3. Results

### 3.1. Results of the Simulation of the Thermal Behaviour of the Virtual Models of the Molar Analysed after Direct Access Cavity Restoration in the Finishing and Polishing Process

The temperature map for this simulation is shown in Figure 11.

For the full thermal evaluation, virtual temperature sensors were placed on the enamel, dentin and dental pulp. The program presents these values in the form of tables of values that can be imported and analysed in Microsoft Excel. 

The temperature evolution in the dental pulp was also monitored (Figure 12).

Figure 13 shows the temperature evolution in the composite, enamel, dentin and pulp, where t42.5 is the time when the temperature of 42.5 is reached. This is determined from the equation of the line, considering the graph T vs. t a line segment for the interval (t1-t2) to which the temperatures (T1-T2) correspond. Parameters a and b belong to the equation of the line and are determined by applying the equation of the line for the known values t1,t2,T1,T2. A linear system of 2 equations with 2 unknowns is formed: a and b.

It can be considered that the temperature chart in the dental pulp (Figure 12) can be approximated by a straight line on small curve segments, as shown in Figure 14.

It is known that the equation of a line can be written as follows:(1)T=a+b·t
where *T* is temperature; *a*, *b* are constant; and *t* is time.

Equation (1) can be written for two times *t*_1_ and *t*_2_ as follows:(2)$$T_{1}=a+b\;\cdot \;t_{1}$$
(3)$$T_{2}=a+b\;\cdot \;t_{2}$$

From Equations (2) and (3), the constants a and b can be extracted:(4)$$b=\frac{T_{2}-T_{1}}{t_{2}-t_{1}}$$
(5)$$a=T_{1}-b\cdot t_{1}$$

For:*T*_1_ = 41.158 °C, *T*_2_ = 43.847 °C, *t*_1_ = 9.7193 s, *t*_2_ = 12.719 sfrom Equations (4) and (5), the constants *a* and *b* are obtained:*a* = 32.4454
*b* = 0.896423

For the temperature of 42.5 °C, you can write:(6)$$T_{42.5}=a+b\;\cdot \;t_{42.5}$$
from where you can determine when the dental pulp reaches the limit value of 42.5 °C, using the equation:(7)$$t_{42.5}=\frac{T_{42.5}-a}{b}$$

Thus, the moment when the dental pulp reaches a temperature of 42.5 °C is: *t*_42.5_ = 11.9375 s.

### 3.2. Results of Simulating the Thermal Behaviour of Virtual Models of the Molar Analysed after Occlusal–Distal Cavity Restoration in the Finishing and Polishing Process

After running the simulation, results in the form of maps and data tables were obtained. Figure 15 shows the temperature map.

Based on the data recorded by the virtual temperature sensors, diagrams were obtained showing the temperature evolution in the enamel, dentin and pulp (Figure 16), and in all the components of the analysed dental system (Figure 17).

Using the mathematical algorithm shown in Equations (1)–(7), it was determined when the dental pulp reaches the temperature limit of 42.5 °C: *t*_42.5_ = 5.099546 s.

### 3.3. Results of Simulating the Thermal Behaviour of Virtual Models of the Analysed Molar after Restoration of the Vertical Slot Cavity in the Finishing and Polishing Process

After running the simulation, result maps and data tables were obtained. Thus, the temperature map is shown in Figure 18.

Based on the data tables obtained from the virtual thermal sensors, temperature diagrams were made of the tooth enamel, dentin, dental pulp (Figure 19) and all components of the system under analysis (Figure 20).

Using the Equations (1)–(7), the time at which the dental pulp reaches 42.5 °C was determined: *t*_42.5_ = 7.545718 s.

### 3.4. Results of the Simulation of the Thermal Behaviour of the Virtual Models of the Analysed Molar after Restoration of the Horizontal Slot Cavity in the Finishing and Polishing Process

After the simulation, which was based on the finite element method, the temperature map was obtained (Figure 21).

Based on the data provided by the virtual thermal sensors, data tables were obtained which, when imported into Microsoft Excel, gave temperature diagrams for dental enamel, dentin, dental pulp (Figure 22) and all components of the analysed system (Figure 23).

Based on the algorithm given by Equations (1)–(7), it was determined when the temperature in the dental pulp reaches 42.5 °C: *t*_42.5_ = 11.2164 s.

The evolution of the dental pulp temperatures during the simulations, i.e., from 0 to 15 s, was analysed. The diagram in Figure 24 shows the temperature evolution compared to the limit temperature of 42.5 °C.

The maximum temperatures obtained in the dental pulp at the end of the 15 s simulation were also evaluated. A comparative diagram for the four types of restoration is shown in Figure 25.

Composite volumes used in the restoration of the upper permanent first molar were also evaluated using virtual environment measurement techniques. A comparative diagram for the four types of filled cavities is shown in Figure 26.

The time limits at which the pulp reaches the temperature of 42.5 °C were also analysed. The comparative diagram of these t42.5 durations is shown in Figure 27.

## 4. Discussions

According to the null hypothesis of the study (h0), all four types of filled cavities equally transmit to the pulp the temperature occurring during the finishing and polishing of the composite. The null hypothesis is rejected by the findings of the study. The heat transfer evidenced in the four types of cavities has an important clinical significance.

Besides aesthetics, the main advantage of composite resins in restoring posterior teeth is the conservative attitude. As a result of chemical bonding to the remaining tooth structures, composite resins no longer require retentive cavities [38]. Increasing emphasis has been placed on minimally invasive preparations, with new cavity designs emerging [6,7]. There are studies that have analysed the effect of cavity design on the fracture resistance of restored teeth [39,40], marginal microleakage [41] and thermal stresses [42].

In the case of mechanical stresses, the results are better documented, as is the case for studies that have looked at marginal closure. This is due to the possibility of carrying out studies under laboratory conditions, simulating various situations encountered in the oral cavity, with the advantage of not raising ethical issues. At the same time, they are less expensive and faster than in vivo studies [43,44].

In the case of thermal studies however, things are a little trickier because of the difficulty of carrying them out, both in vivo and in vitro.

In 1965, Zach and Cohen [45] reported that an increase in intrapulpal temperature in monkey teeth, by 5.5 °C and 11 °C intrapulpally for 10 s, respectively, resulted in pulpal necrosis rates of 15% and 60%, respectively, at a three-month examination. Increasing the temperature by 16 °C produced 100% irreversible pulpal damage. There have been other studies that have attempted to determine the critical pulp temperature [46,47], but the 5.5 °C threshold has remained a standard for subsequent studies to this day.

Most thermal studies in restorative dentistry have focused on the effect of light curing in heat production [48,49,50].

However, another important source of heat generation is the finishing and polishing of composite resin fillings. Essential during clinical restorative procedures to enhance the aesthetics and longevity of filled teeth [51], they involve, by their nature, friction between finishing burs, abrasive paper discs or rubber cups and the surfaces to be polished, and can generate enough heat to irreversibly damage the pulp [52].

According to the study by C. M. Hannah et al., the surface temperature of the composite subjected to finishing and polishing without cooling is at least 140 °C [22]. Obviously, this problem could be prevented by using cooling during this stage. In order to have the best finishing and polishing technique, it is vital to know not only whether or not irrigation improves the quality of the final result, but also whether it alters the material properties or damages the remaining hard structures [51].

The abundant use of coolant during finishing and polishing procedures is considered a simple and effective method of pulp protection. Currently, the use of a flow rate between 30 and 50 mL/min is considered the cooling standard during dental treatments [53]. However, the cooling-water temperature may affect the intrapulpal temperature. Although lower water temperature has a higher efficiency in heat absorption [54], clinical use is limited by the risk of affecting pulpal blood flow by reducing it when the temperature drops below 31 °C [55]. Besides the disadvantages to the dental pulp of the heat produced by finishing and polishing without cooling however, studies also mention some advantages. When finishing and polishing are carried out without cooling, due to the occurrence of high-temperature friction, the outer composite layer may undergo melting of the surface particles. This results in a smoother surface [56].

Nasoohi et al. found, in their study, an increase in the hardness of all composite samples analysed when finishing was performed without cooling [57].

As finishing and polishing without cooling continue to be used by practitioners, the limits to which they are safe for maintaining dental pulp homeostasis should be known. In this study, which examined the influence of filling volume and design on heat transmission to the pulp generated during finishing and polishing without cooling, it can be seen that the volume of the composite filling clearly influences the results.

In the case of the tooth restored with an occlusal–distal filling, which quantifies 29.35 mm^3^ of the total tooth volume, the intrapulpal temperature recorded was approximately 3.5 °C higher than in the other three situations, after a 5 s exposure at 140 °C. In the case of the other three fillings (horizontal slot, vertical slot and direct access), the results were similar. It can thus be seen that, for all four situations, finishing and polishing without cooling for 5 s is safe for maintaining pulpal health; although, in the case of the occlusal–proximal filling, the intrapulpal temperature (41.39 °C) was at the upper limit of the threshold for irreversible damage to the dental pulp (42.5 °C).

Analysing the literature, we found that there are no studies that specifically examine the effect of dental filling volume on heat transfer to the pulp chamber, and research on dental materials and techniques often addresses this issue indirectly. Dental researchers frequently investigate factors related to the properties of filling materials [58], cavity preparation techniques and their impact on tooth structure and pulp vitality.

The fact that the volume of the filling influences the heat transfer to the pulp can be explained, from a physical point of view, in several ways. Based on the idea that every body has a thermal mass, defined as the ability of a material to absorb, store and release heat [59], we can deduce by analogy that a larger volume of composite will have more thermal mass, which means it can absorb, retain and conduct more heat during finishing and polishing. This may increase the likelihood of easy heat transfer to the pulp.

For the exposure duration and intensity of the causal factor, as the temperature was constant for all simulations (140 degrees), time remained the only variable, which highlighted the influence of the volume of the filling on the results obtained. The critical temperature (42.5 °C) was reached almost twice as fast (5.099 s) in the case of the occlusal–distal obturation compared to the other three obturations: horizontal slot, vertical slot and direct access. There, the critical temperature was reached after 11.2162 s, 7.5457 s and 11.9375 s, respectively.

One aspect that should not be neglected at all is the attitude towards healthy hard structures. Singh A. et al. demonstrated the influence of the amount of remaining crown in increasing the intrapulpal temperature; since the amount of remaining hard tissue decreases, the volume of filling requiring light-curing, finishing and polishing increases [60].

In this study, starting from the simulation of a caries process on the distal aspect of an upper primary molar, and digitally creating all four cavity types so that the entire caries was embedded in the cavity, the most conservative was the box cavity (12.2 mm^3^) and the most invasive, the occlusal–proximal cavity (29.35 mm^3^). Between the horizontal slot and vertical slot cavity, the volume difference was less than 1 mm^3^. However, in the case of the latter two situations, while the intrapulpal temperatures recorded were similar after 5 s, reaching the critical temperature of 42.5 °C occurred after a considerable time difference (11.2164 s for the horizontal slot and 7.5457 s for the vertical slot). A possible explanation for this phenomenon could be that, in addition to the volume of composite driving the temperature, an equally large influence is the area subjected to finishing and polishing.

Analysing the specialized literature, we found that the few studies that studied the influence of filling volume on the transmission of heat to the dental pulp obtained similar results.

In a study by Janeczek M. et al., which aimed to comparatively evaluate the temperature occurring during the light-curing of several selected composites and to analyse the influence of their volume on the thermal parameters, it was found that, for all materials analysed, the maximum temperature was correlated with the maximum volume of the samples. Thus, the small samples, which measured approximately 2 × 4 mm, did not increase the intrapulpal temperature above 42.5 °C for any of the materials examined. The most significant temperature increase, over 42.5 °C, was found for samples weighing over 60 mg (2 × 8 mm) [59]. In the few studies conducted on this topic, P. Mirzakoucheki Boroujeni et al. investigated the influence of finishing (wet and with cooling) on the heat generation of 9 mm diameter composite samples with three thicknesses (2, 3 and 4 mm). The results showed that the maximum exterior–interior surface temperatures of the composite were recorded for samples of a size of 9 × 4 mm, where continuous dry finishing and polishing was performed. The lowest temperatures were recorded for the samples of a size of 9 × 2 mm, finished and polished with continuous cooling. The maximum temperature was recorded for dry–continuous finishing and was 53.7 °C [61].

Regarding the maximum temperature recorded after the 15 s of simulation, for the four situations it was found that the influence of the volume of the obturation on the transmission of heat to the pulp was no longer so obvious. Although, in the case of the occluso–distal obturation, the highest intrapulpal temperature was recorded (61.74 °C), and between the direct access and horizontal slot obturations, the values were similar (45.89 °C and 44.28 °C, respectively). An intermediate value was obtained for the vertical slot obturation (52.6 °C). This could be explained by the ability of dental hard tissues to disperse heat when the irritating factor acts for a longer period of time.

It is important to mention that this study has a number of limitations due to the fact that dental tissues were considered homogeneous, thus ignoring the specific morphology of dentin. Enamel is a good conductor of temperature, and the role of an insulator is played by dentin. However, dentin has different thermal conductivity in relation to its distance from the pulp chamber. The smaller the remaining dentin layer, the larger the diameter of the dentinal canaliculi and the higher its thermal conductivity [62]. Another limitation of the study is the fact that specialized literature contains few similar studies for carrying out a detailed comparative analysis.

## 5. Conclusions

Analysing the diagram of the time limits at which the dental pulp reaches the critical temperature of 42.5 °C, it can be seen that the shortest time limit occurs in occlusal–distal restorations and the longest in direct access restorations.

It can be seen that the volume of composite is very likely to almost inversely proportionally influence the time at which the dental pulp reaches a temperature of 42.5 °C. Thus, in the case of vertical and horizontal slot cavities, although the difference in volume is small, in terms of heat transfer behaviour, the horizontal slot cavity is preferred over the vertical slot cavity.

This study is significant because it will provide clinicians with recommendations on the preparation of an optimal cavity type, in the case of a proximal carious process, adapted to the existing clinical situation in order to minimize pulpal thermal injury during filling finishing and polishing.

A working protocol can be issued so that finishing and polishing without cooling is safe for pulpal health. For occlusal–distal restorations, finishing and polishing should be limited to a duration of less than 5 s, followed by a break of at least 15 s. For direct access and horizontal slot restorations, finishing times of 11 s or less are recommended. For vertical slot restorations, finishing times of 6 s are recommended.

## Figures and Tables

**Figure 1 jfb-15-00086-f001:**
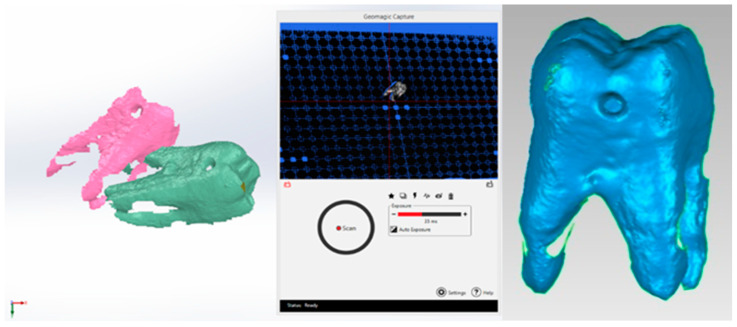
Specific operations of three-dimensional molar scanning.

**Figure 2 jfb-15-00086-f002:**
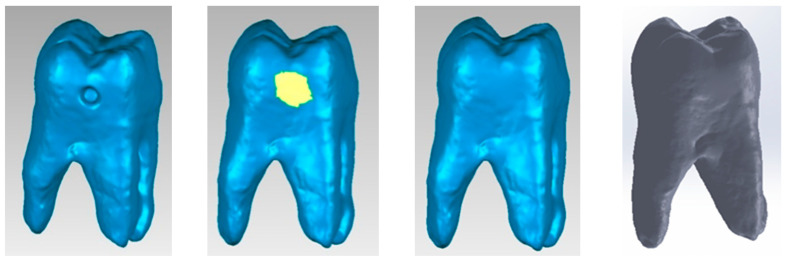
Obtaining the integral molar model.

**Figure 3 jfb-15-00086-f003:**
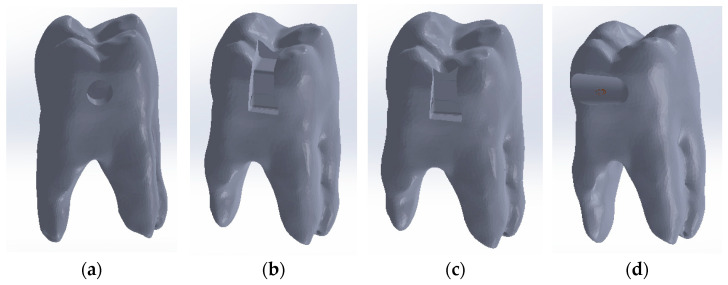
(**a**) Direct access cavity; (**b**) distal–occlusal cavity; (**c**) vertical slot cavity; (**d**) horizontal slot cavity.

**Figure 4 jfb-15-00086-f004:**
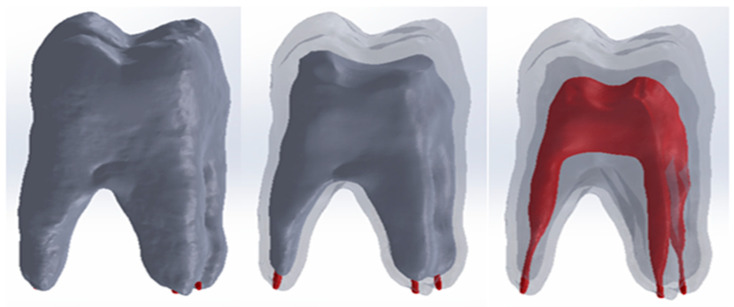
Alignment of the three models (views with different degrees of transparency).

**Figure 5 jfb-15-00086-f005:**
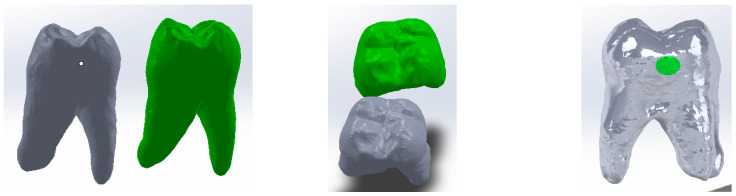
The two models in SolidWorks.

**Figure 6 jfb-15-00086-f006:**
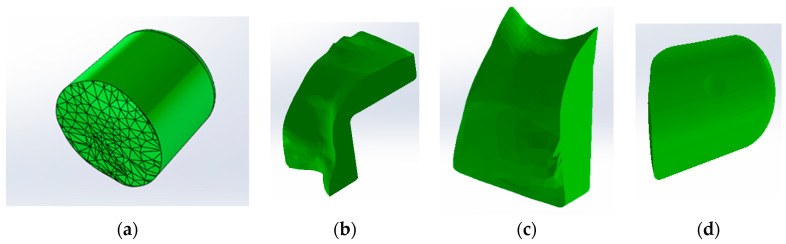
(**a**) Composite-filling model in direct access cavity situation; (**b**) model of composite filling for the occlusal–distal cavity; (**c**) model of the composite layer for the vertical slot cavity; (**d**) composite layer model for the horizontal slot cavity.

**Figure 7 jfb-15-00086-f007:**
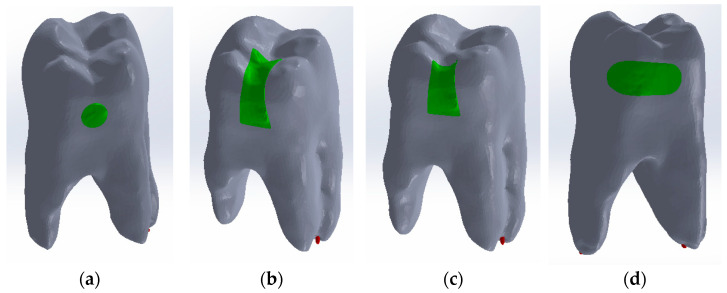
Model of the filled molar for: (**a**) direct access cavity; (**b**) occlusal–distal cavity; (**c**) vertical slot cavity; (**d**) horizontal slot cavity.

**Figure 8 jfb-15-00086-f008:**
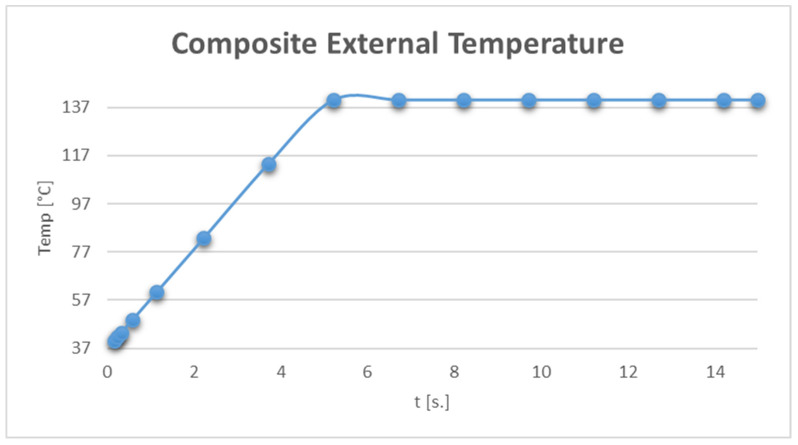
Temperature evolution during the virtual experiment.

**Figure 9 jfb-15-00086-f009:**
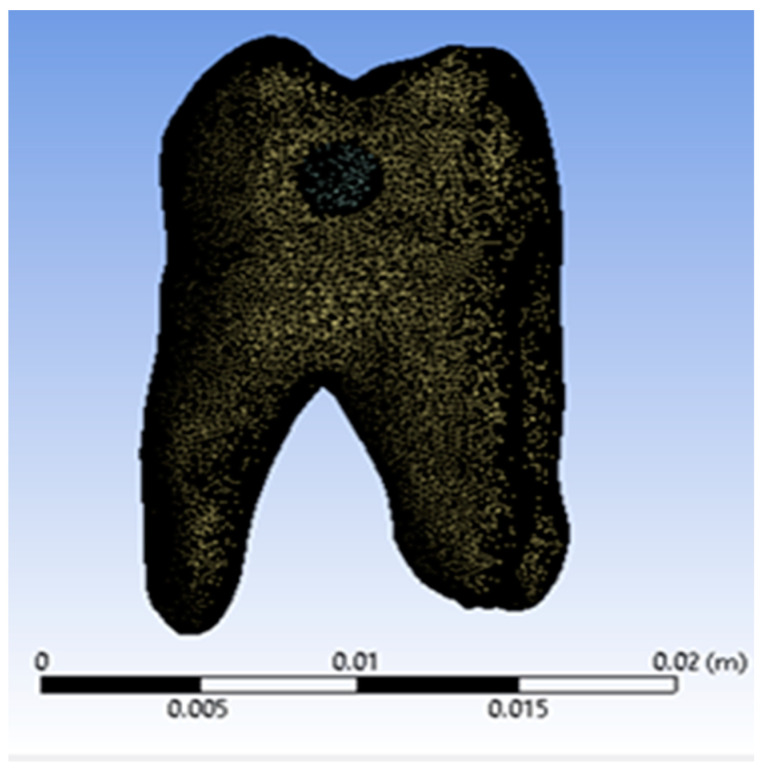
Finite element structure of the analysed system.

**Figure 10 jfb-15-00086-f010:**
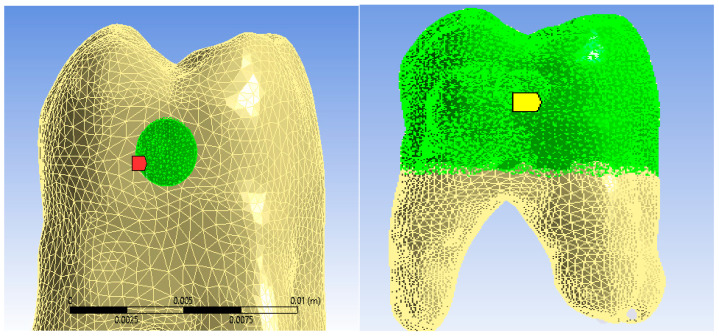
The surface that was subjected to the temperature source, and the surface to which convection was applied.

**Figure 11 jfb-15-00086-f011:**
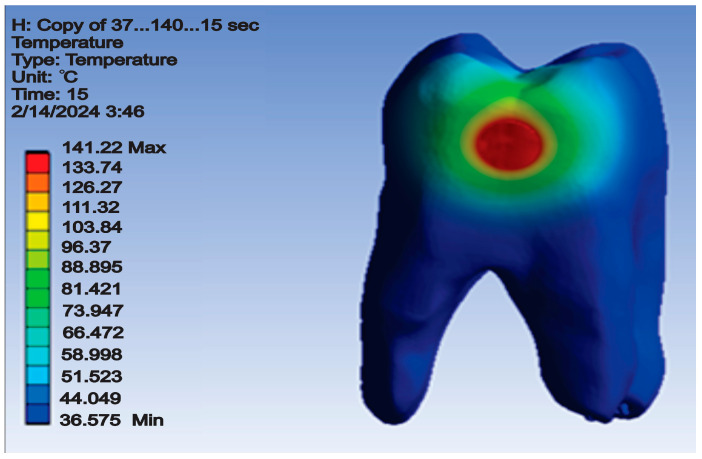
Temperature map for direct access cavity restoration.

**Figure 12 jfb-15-00086-f012:**
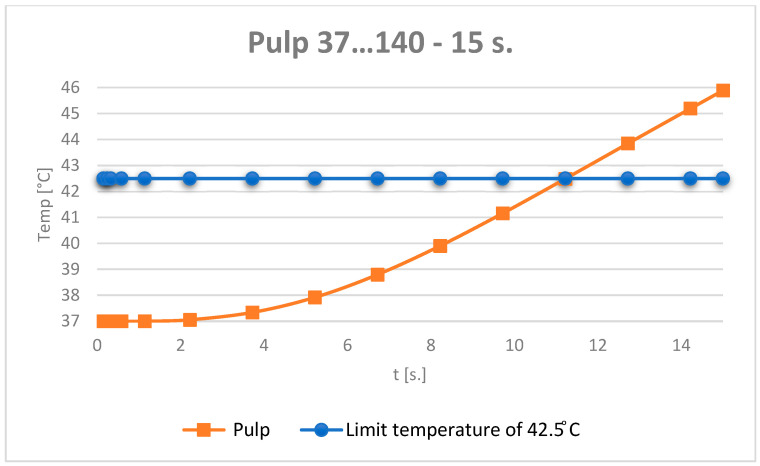
Temperature evolution in the dental pulp and limit temperature of 42.5 °C.

**Figure 13 jfb-15-00086-f013:**
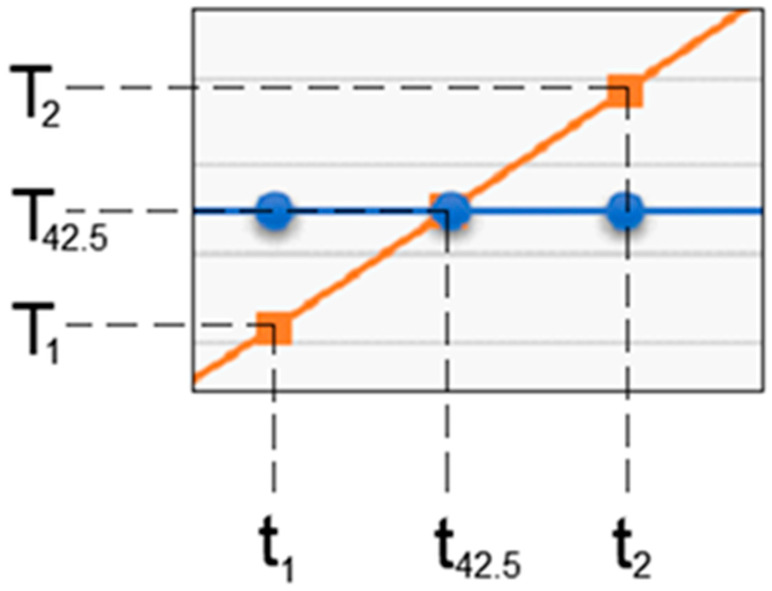
Approximating the curve with a straight line. Temperature evolution in the dental pulp (coloured in orange) and limit temperature of 42.5 °C (coloured in blue).

**Figure 14 jfb-15-00086-f014:**
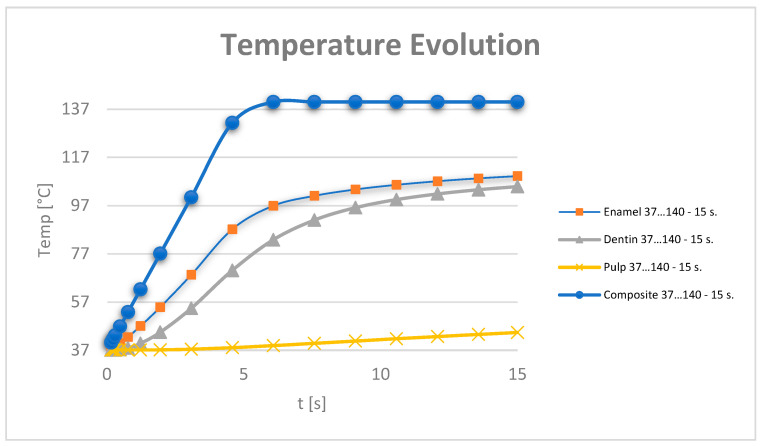
Temperature evolution in dental structures.

**Figure 15 jfb-15-00086-f015:**
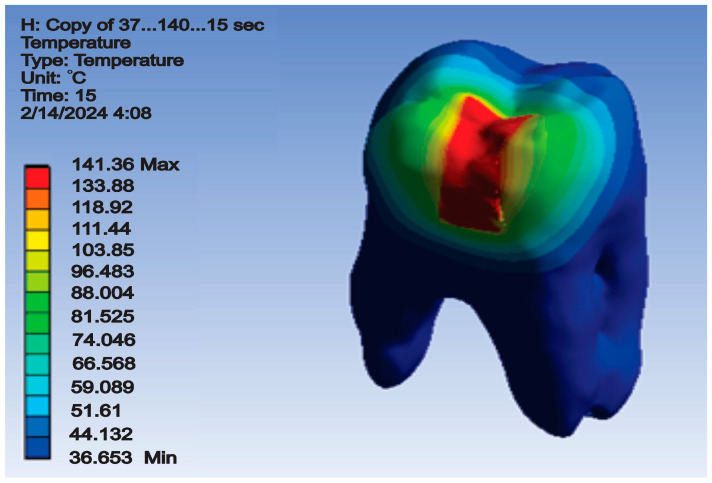
Temperature map for occlusal–distal cavity restoration.

**Figure 16 jfb-15-00086-f016:**
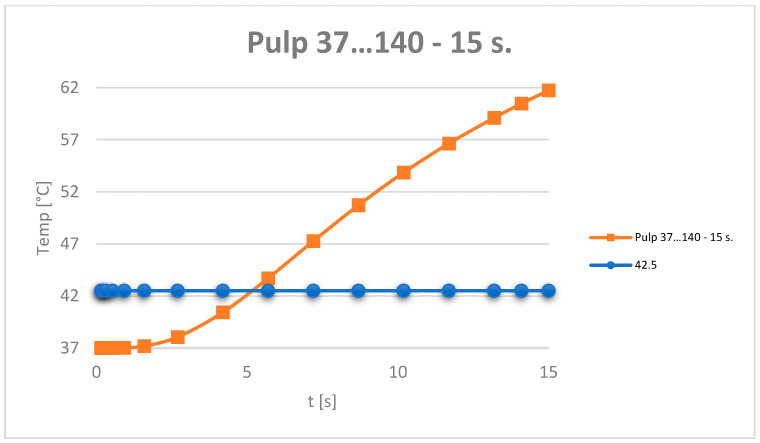
Temperature evolution in the dental pulp.

**Figure 17 jfb-15-00086-f017:**
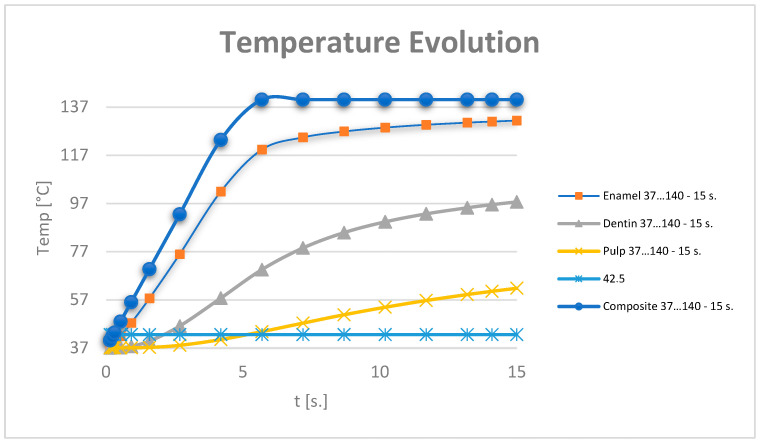
Temperature evolution in the components of the analysed dental system.

**Figure 18 jfb-15-00086-f018:**
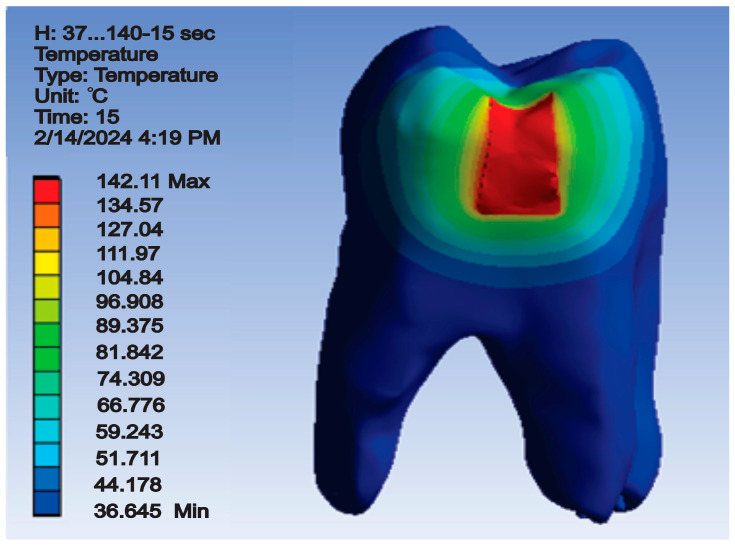
Temperature map for vertical slot cavity restoration.

**Figure 19 jfb-15-00086-f019:**
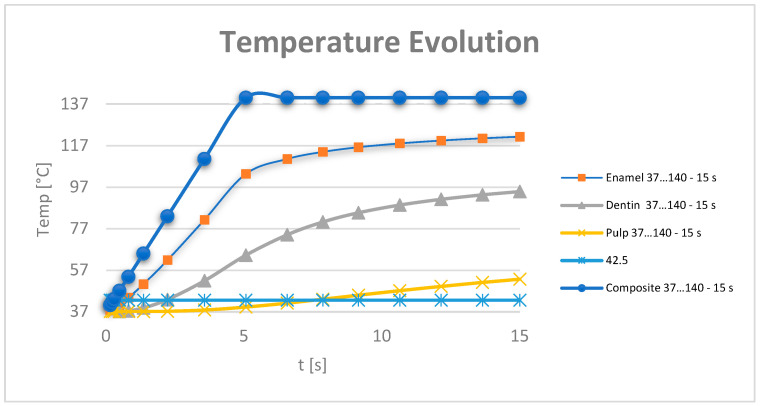
Temperature evolution in the dental pulp.

**Figure 20 jfb-15-00086-f020:**
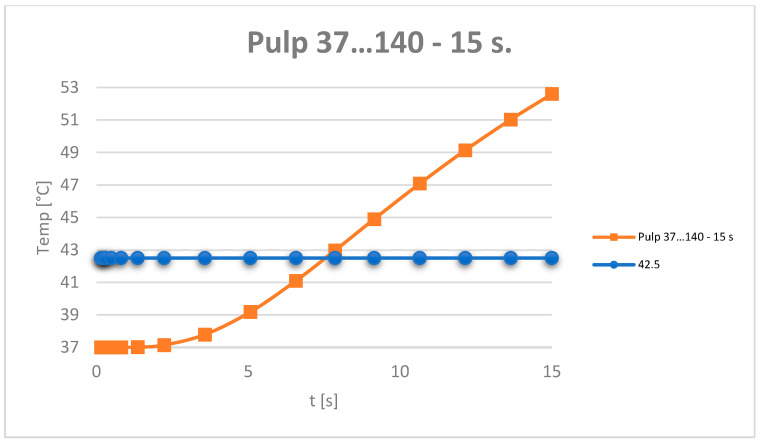
Temperature evolution in the components of the analysed dental system.

**Figure 21 jfb-15-00086-f021:**
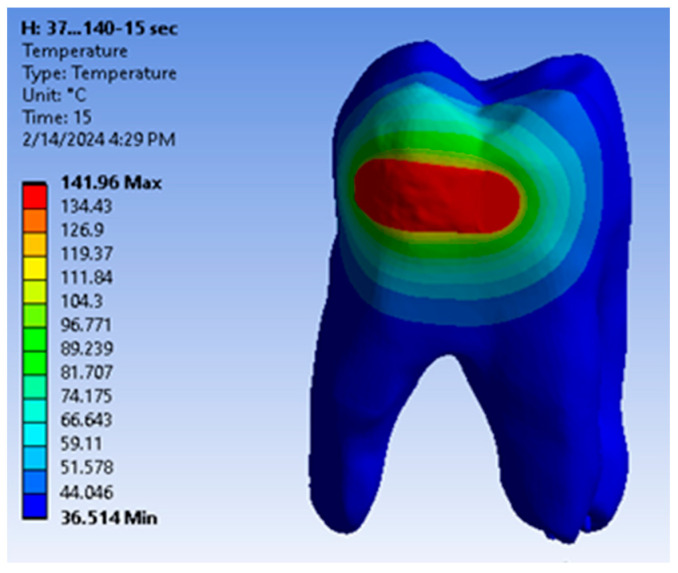
Temperature map for horizontal slot cavity restoration.

**Figure 22 jfb-15-00086-f022:**
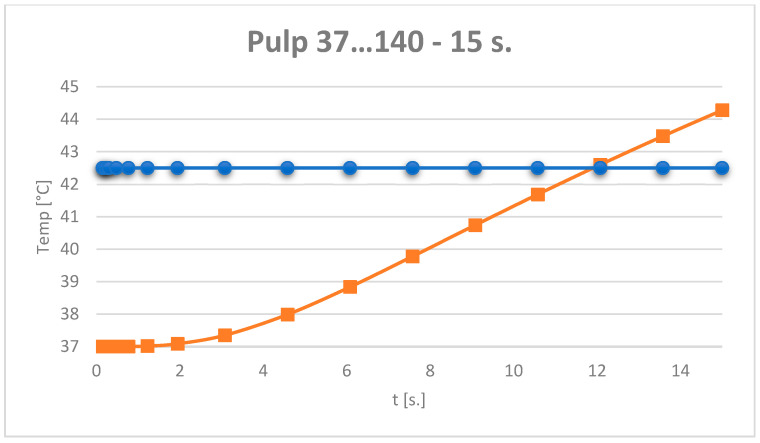
Temperature evolution in the dental pulp.

**Figure 23 jfb-15-00086-f023:**
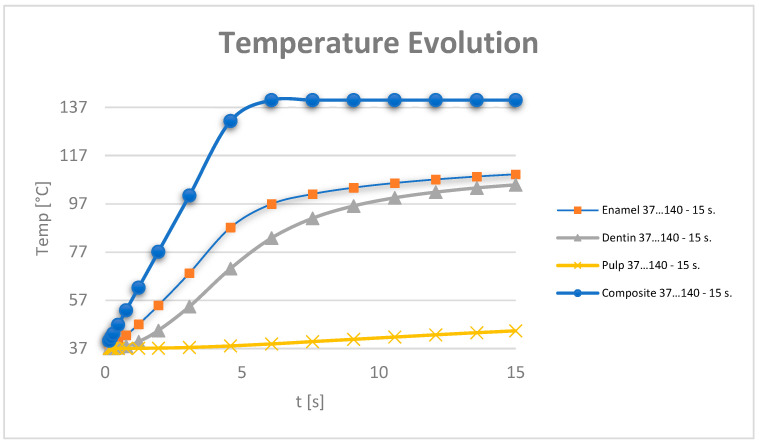
Temperature evolution in the analysed system components.

**Figure 24 jfb-15-00086-f024:**
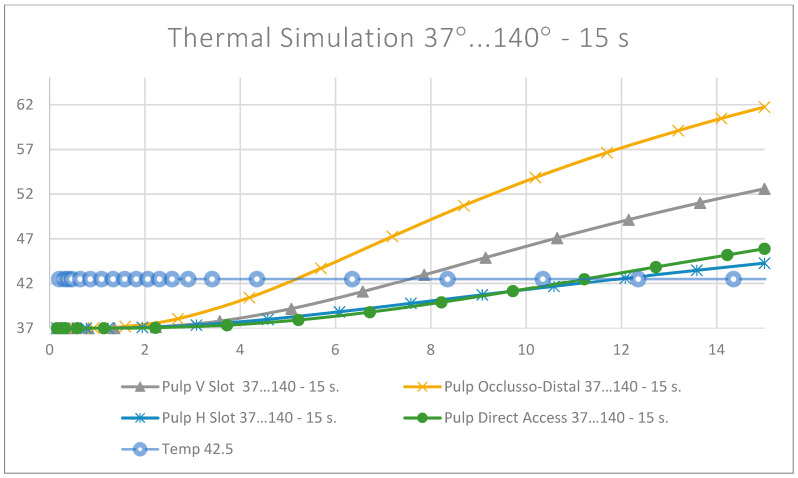
Evolution of the temperatures in the dental pulp.

**Figure 25 jfb-15-00086-f025:**
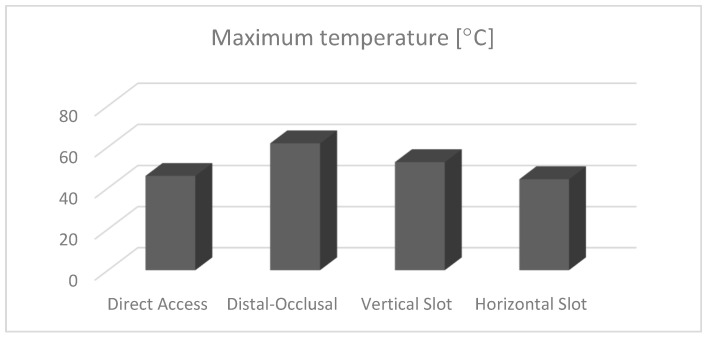
Maximum temperatures in the dental pulp.

**Figure 26 jfb-15-00086-f026:**
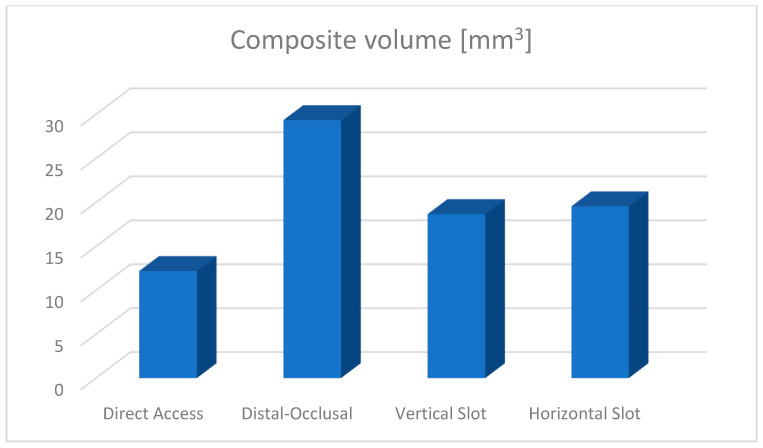
Diagram of composite volumes.

**Figure 27 jfb-15-00086-f027:**
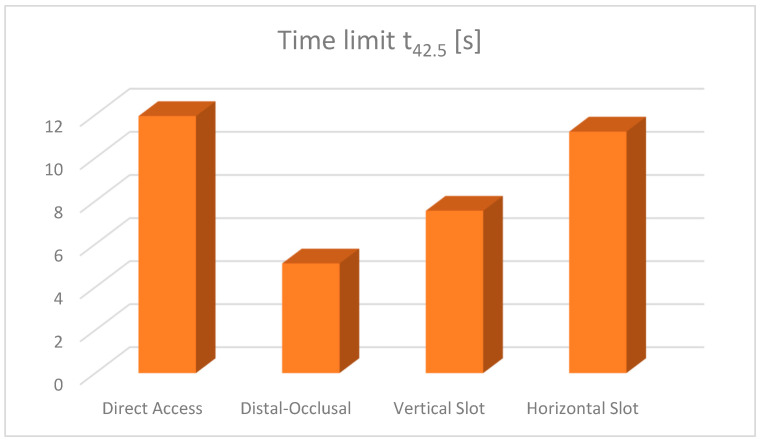
Time limit diagram.

**Table 1 jfb-15-00086-t001:** Physico-thermal properties of the simulation components.

Component	Young’s Modulus [GPa]	Poisson’s Ratio	Density [kg/m^3^]	Thermal Conductivity [W/m·K]	Specific Heat [J/g × K]
Enamel	80	0.33	2.800	0.84	750
Dentin	20	0.31	2.000	0.36	1.302
Pulp	0.003	0.45	1.000	0.0418	4200
Filtek Supreme XT	5.76	0.45	1.500	1.18	1.37

## Data Availability

The authors declare that the data from this research are available from the corresponding authors upon reasonable request.

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
