# Peer review of "A Finite Element Method Study on a Simulation of the Thermal Behaviour of Four Methods for the Restoration of Class II Cavities"

_jfb, 2024, doi:10.3390/jfb15040086_

Round 1

Reviewer 1 Report

Comments and Suggestions for Authors

Dear authors, thank you for contributing with this interesting in vitro research work. The article is well structured, however there are some issues that should be addressed before the work can be fulluy considered for publication. Please reduce the number of graphs and figures; it is difficult to understand and may reduce interest for the readers.

The article is too much long (especially the results and discussion sections), please try to concentrate the main concepts.

Author Response

From the authors:

Thank you for the comments received!

We are grateful for the attention and effort spent in reviewing our work, and valuable comments made by the respectful editor and reviewers.

We are sending you detailed answers to your comments below:

No

Remarks

Author's response

1

This study is significant because it will provide clinicians with recommendations on 105 the preparation of an optimal cavity type in the case of a proximal carious process, adapted to the existing clinical situation, in order to minimize pulpal thermal injury during filling finishing and polishing. “ This sentence should be incorporated in the discussion/conclusion sections, not in the introduction.

The mention phrase was relocated within the Conclusions chapter, on line 541.

2

Moreover, the authors list the “filling" procedure in the aims section of the study, but there is no mention in the mat&met (the authors report only the results of finishing and polishing procedures). They should remove it from the introduction or incorporate in the results (no additional graphs!

The term “filling" was used in the aim of the study, but also in the material and method section, where the type of material used was specified (line 110), together with its physical-mechanical properties (line 241). Moreover, in subchapter 2.6.1, a thermal protocol was proposed, which strictly targets the surface of the obturation material, considering that finishing and polishing are done at this level. Taking into account that the main purpose of the study was to establish a finishing and polishing protocol, which would be safe for the integrity of the pulp, in the case of obturation of the 4 types of cavities with the filling material Filtek Supreme XT, but also to what extent the filling volume influences the results , we did not analyze in detail other aspects related to the material used.

3

Please reduce the number of graphs and figures (MAX 20); it is difficult to understand and may reduce interest for the readers.

The number of graphics and figures has been reduced so that the content is more concise.

4

The description of the virtual model needs to be summarized.

The description of the virtual model has been revised, by shortening the mentioned description.

5

The article is too much long (especially the results and discussion sections), please try to concentrate the main concepts

The mentioned chapters have been revised so that the main aspects are presented more clearly.

6

Emphasize the conclusion on the possible protocols to avoid heat damage to the pulp during finishing/polishing

We reviewed the conclusions regarding the protocol as indicated.

Reviewer 2 Report

Comments and Suggestions for Authors

The study of posterior tooth restoration using composite resin is a topic of special concern. Due to Lack of clear understanding of the actual heat transfer during dental surgery and difficulty in analyzing thermal parameters in vivo, in this manuscript, using the finite element method, the simulation of thermal behavior of four methods of restoration of class II cavities was investigated. This study is significant to provide clinicians with commendations on the preparation of an optimal cavity type in the case of a proximal carious process, to minimize pulpal thermal injury during filling finishing and polishing. In my opinion, the following comments should be considered.

1. In the discussion section, the focus should be on discussing the results obtained, especially in the differences and commonalities as well as their reasons of the relevant results shown in Figure 21-34. At the same time, combining with literature reports, the consistency or differences between the obtained results and literature reports should be discussed. The statements in the present discuss section is too generalized and not very close to the results obtained.

2. the experimental section, it is necessary to explain the basic principles of how material property parameters are applied to calculations. The influence of the parameters used in the calculation on the obtained results should also be explained to facilitate readers' understanding of the obtained calculation results. How to obtain Figure 11 should be explained.

Author Response

From the authors:

Thank you for the comments received!

We are grateful for the attention and effort spent in reviewing our work, and valuable comments made by the respectful editor and reviewers.

We are sending you detailed answers to your comments below:

No

Remarks

Author's response

1

In the discussion section, the focus should be on discussing the results obtained, especially in the differences and commonalities as well as their reasons of the relevant results shown in Figure 21-34. At the same time, combining with literature reports, the consistency or differences between the obtained results and literature reports should be discussed. The statements in the present discuss section is too generalized and not very close to the results obtained.

The discussion section has been revised so that the relevant differences and common points are better highlighted (line 515).

The mentioned generalization is precisely caused by the lack of relevant reports from the specialized literature. The few studies mentioned in the Discussion chapter obtained similar results.

The impossibility of reporting in detail the results obtained is a limitation of the present study.

4

the experimental section, it is necessary to explain the basic principles of how material property parameters are applied to calculations. The influence of the parameters used in the calculation on the obtained results should also be explained to facilitate readers' understanding of the obtained calculation results. How to obtain Figure 11 should be explained.

Additions were made regarding the basic principles of how material property parameters are applied to calculations (line 135).

The figure 11 (curently 8) was explained betwen lines 224 -227.

Reviewer 3 Report

Comments and Suggestions for Authors

1. There are too many pictures, it is recommended that the author merge the pictures with similar content

2. Curves in Figure 19 and Figure 20 are not marked

3. The curve in Figure 31 is too vague

4. The sequence of FIG. 32.33.34 is not arranged according to the sequence of experimental method design.

a is directly adjacent to the surface hole, b is adjacent to the surface hole, c is adjacent to the vertical hole, d is adjacent to the horizontal hole

Author Response

From the authors:

Thank you for the comments received!

We are grateful for the attention and effort spent in reviewing our work, and valuable comments made by the respectful editor and reviewers.

We are sending you detailed answers to your comments below:

No

Remarks

Author's response

1

There are too many pictures, it is recommended that the author merge the pictures with similar content

The number of images has been reduced so that the content is more concise.

2

Curves in Figure 19 and Figure 20 are not marked

The marking corresponding to the 2 figures has been revised.

3

The curve in Figure 31 is too vague

The graphic has been revised so that the content is as clear as possible.

4

The sequence of FIG. 32.33.34 is not arranged according to the sequence of experimental method design a is directly adjacent to the surface hole, b is adjacent to the surface hole, c is adjacent to the vertical hole, d is adjacent to the horizontal hole

The sequence was reordered so that it is consistent with the stages of the experimental method.

Reviewer 4 Report

Comments and Suggestions for Authors

I don't have any remarks to this paper.

Author Response

From the authors:

Thank you for the comments received!

We are grateful for the attention and effort spent in reviewing our work.